# Assessment of the Spatial Invasion Risk of Intentionally Introduced Alien Plant Species (IIAPS) under Environmental Change in South Korea

**DOI:** 10.3390/biology10111169

**Published:** 2021-11-12

**Authors:** Pradeep Adhikari, Yong-Ho Lee, Yong-Soon Park, Sun-Hee Hong

**Affiliations:** 1Institute of Ecological Phytochemistry, Hankyong National University, Anseong 17579, Korea; pdp2042@gmail.com (P.A.); yongho@korea.ac.kr (Y.-H.L.); 2OJeong Resilience Institute, Korea University, Seoul 02841, Korea; 3Biotechnology Research Institute, College of Natural Sciences, Chungbuk National University, Cheongju 28644, Korea; 4School of Plant Science and Landscape Architecture, College of Agriculture and Life Sciences, Hankyong National University, Anseong 17579, Korea

**Keywords:** climate change, intentionally introduced alien plants, invasion risk, land cover change, province, spatial distribution

## Abstract

**Simple Summary:**

The spatial distribution and invasion risk of 10 intentionally introduced alien plant species (IIAPS) in South Korea were predicted from a species distribution model via a maximum entropy modeling approach. According to the model predictions, future environmental changes are likely to enlarge the range of the spatial distribution of all studied IIAPS in South Korea except *Medicago sativa.* We classified the IIAPS into three categories based on their spatial distribution and potential to spread; this revealed that four species (*Coreopsis lanceolate**, Eragrostis curvula, Ageratina altissima*, and *Lolium perenne*) have the highest potential for invasion. Moreover, we classified invasion risk into three categories, low, moderate, and high, and estimated the area in each category. We found that, under current conditions, much of the country is at low risk (47.96%) of invasion, but by 2050 >54% of the country’s total area will be at high risk of invasion by IIAPS. Serious invasion of IIAPS into cropland, pastures, and forests results in the loss of native biodiversity and damage to the national economy. Therefore, immediate action is required to control and manage IIAPS in South Korea.

**Abstract:**

Predicting the regions at risk of invasion from IIAPS is an integral horizon-scanning activity that plays a crucial role in preventing, controlling, and eradicating invasive species. Here, we quantify the spatial distribution area and invasion risk of IIAPS using a species distribution model under different levels of environmental change in South Korea. From the model predictions, the current average spatial extent of the 10 IIAPS is 33,948 km^2^, and the individual spatial extents are estimated to change by −7% to 150% by 2050 and by −9% to 156% by 2070. The spatial invasion risk assessment shows that, currently, moderate-to-high invasion risk is limited to coastal areas and densely populated metropolitan cities (e.g., Seoul, Busan, and Gwangju), but that the area with this level of risk is expected to spread toward the central and northern regions of the country in the future, covering 86.21% of the total area of the country by 2070. These results demonstrate that the risk of invasion by IIAPS is estimated to enlarge across the whole country under future environmental changes. The modeling system provided in this study may contribute to the initial control and strategic management of IIAPS to maintain the dynamic ecosystems of South Korea.

## 1. Introduction

Invasion risk is the likelihood that non-native species will be introduced to and become established within a novel ecosystem, either intentionally or inadvertently, thereby threatening native biodiversity, ecosystem services, and human well-being [1,2,3]. Some alien plant species are intentionally introduced to non-native regions because of their importance in agriculture, horticulture, pastoral productivity, or land rehabilitation [4,5]. After the successful establishment and proliferation of introduced species, many become dominant in diverse areas with subsequent invasion of non-target ecosystems along various pathways [6], ultimately resulting in unexpected conservation challenges [5].

Usually, the non-native plant species have some specific functional traits [7,8] that support the introduction, naturalization, and invasion of new areas [9]. They are often ruderal, growing along transportation corridors, irrigation canals, the seashore, and riversides as noxious weeds [10]. Increasing the number of non-native species in some regions amplifies the magnitude of invasion risk [11]. Uncontrolled expansion of such species alters the pools and fluxes of an ecosystem and can cause grievous reductions in crop yields, resulting in substantial economic losses [2,12,13,14,15].

Global climate change may exacerbate the risk of invasion of alien plant species through ecosystem destruction and increased competition within native ecological systems because of elevated carbon dioxide (CO_2_) [16,17]. In addition, climate change encourages rapid range shift and changes the life cycle, life-history traits, and population dynamics of invasive plants [18,19,20,21]. In the last century, the global temperature increased by 0.78 °C and is projected to increase by 2.6 °C to 4.8 °C by 2100 [22]. In addition to climate change, changes in land cover are important to the introduction, establishment, and spread of alien plant species, including invasive weeds [23]. Land cover changes may provide a suitable habitat for particular plants to invade an area, inhibiting others and acting as a dispersal corridor [24,25,26]. Land cover changes such as forest clearing for urbanization, agriculture, or pastoral purposes facilitate biological invasion [26].

Alien plant species were first introduced into South Korea prior to 1949 for agricultural or horticultural reasons, pastoral land management, or erosion control [27]. Eighty-one taxa of alien plants introduced into South Korea have successfully naturalized and seriously threatened native plant communities growing in fields, orchards, forests, and pastures [28,29]. Jeju province has the highest number of alien plant species in South Korea [27,28]. It has been reported that invasive and alien plant species cause approximately 19.6 million USD of economic damage per year in South Korea, and the government invests about 4.3 million USD per year in the management of invasive weeds, including control and eradication [15]. Moreover, South Korea’s land cover has altered rapidly due to the development of the modern transportation system, industrialization, and coastal land reclamation [30,31]. Under these circumstances, the issue of how to monitor and manage intentionally introduced alien plant species (IIAPS) in South Korea should be an important subject of discussion.

A maximum entropy (MaxEnt) modeling approach is normally required to assess the spatial distribution of IIAPS and estimate their potential invasion risk under climate and land cover changes [32,33,34,35]. Although MaxEnt is widely regarded as a suitable tool, almost no MaxEnt studies on the spatial distribution of IIAPS have been reported in South Korea: we designed this study to bridge this gap. The main objectives of this study were (1) to analyze the impacts of bioclimatic, land cover change, road transportation, and water proximity variables on the occurrence probability of IIAPS; (2) to evaluate the spatial distribution of IIAPS with the MaxEnt algorithm and estimate the risk of spatial invasion under current and future environmental changes in South Korea; and (3) to classify the IIAPS into groups according to their spatial distribution and invasion potential, to prioritize control and management strategies. We estimated three categories of spatial invasion risk—low, moderate, and high—across the country and existing provinces. Our study shows that future environmental changes increased both the extent and the intensity of spatial invasion of IIAPS in South Korea and that the studied species could disrupt ecosystems on a large scale in the future. The modeling system used sheds light on the potential spatial distribution of invasive weeds and IIAPS in the near future and the associated risk assessments.

## 2. Materials and Methods

### 2.1. Study Areas

South Korea is located in East Asia and constitutes the southern portion of the Korean Peninsula. South Korea has a total landmass of 98,477 km^2^, excluding the demilitarized zone along the boundary with North Korea (Figure 1). It is surrounded by three seas along the east (East Sea), west (Yellow Sea), and south (East China Sea) and has a coastline approximately 2413 km in length [36]. The country is mostly mountainous (approximately 70% of the country) in the east and north, with lowlands and flat plains occupying the remaining 30% in the west and south [36].

The climate of South Korea can be mainly divided into cold-temperate, temperate, and warm-temperate regions, located in the northern region plus high mountains, the central region, and the southern region, respectively, with four distinct seasons of spring, summer, autumn, and winter [37]. The winter is long, dry, and cold, while the summer is short, humid, and hot. The autumn and spring are sunny and usually dry and pleasant. The average winter temperatures range between −6 °C and 3 °C, and the average summer temperatures range between 23 °C and 26 °C. Similarly, annual precipitation records 1000 to 1800 mm [38]. The southern coast, adjacent mountains, and Jeju Island have the largest amount of rainfall, recording over 1500 mm in a year [38].

The vegetation of South Korea is primarily divided into alpine, subalpine, coniferous, deciduous broadleaf, and temperate broadleaf. The overall biodiversity has been reported to comprise 41,483 species, of which 2177 are endemic [36,37]. Of the total species, 5308 are vascular plants, 22,612 are invertebrates, and 1899 are vertebrates [36]. The current study focused on 250 districts in 17 administration divisions (provinces and metropolitan cities) in South Korea (Figure 1).

### 2.2. Occurrence of Intentionally Introduced Alien Plant Species

Ten alien plant species (Table 1 and Figure 2) introduced intentionally in South Korea were selected based on their rapid range expansion, their degree of invasion into natural ecosystems [27,28], and the availability of minimum species occurrence records. Species presence points for such species were noted through field surveys performed from March 2014 to November 2020, and additional presence points were collected from the literature [28,39]. We used Garmin GPSmap 64SX (Garmin, Ltd., Seoul, Korea) to record the occurrence and location of species to measure the dispersal of invasive weeds. The species presence survey, plot design, and survey techniques were all conducted according to the National Institute of Ecology [28]. Multiple species presence points in the same grid at a spatial resolution of 1 km^2^ were erased and then retained as a single unique point per grid by applying the spatially rarefy occurrence data tool using the SDM toolbox 2.4 in the Arc GIS [40]. This avoids incorrect inflation of model outputs and overfitting because of the spatial autocorrelation [41]. The total species occurrence points for the 10 IIAPS were reduced from 7389 to 4671 after spatial filtering, and those points were employed in the MaxEnt modeling. The species occurrence for each IIAPS is presented in the Appendix A.

### 2.3. Environmental Variables

We collected raw data for bioclimatic variables [42,43], including monthly minimum and maximum temperatures and monthly precipitation, from the Korea Meteorological Administration (KMA). We estimated climate change scenarios according to the representative concentration pathways (RCPs) of 4.5 and 8.5 in 2050 and 2070, respectively, which suggest that the projected global mean surface temperature will increase by 1.4–1.8 °C and 2.0–3.7 °C, respectively, compared with current levels [22].

The physical properties of the earth’s surface, atmosphere, cryosphere, and ocean are all considered in global circulation models (GCMs). These models keep track of the basic systems that adapt to climate change (e.g., surface albedo changes, aerosols, solar irradiance, and concentrations of greenhouse gases) [44]. The HadGEM3-RA is a regional atmospheric model developed by the Meteorological Office Hadley Center (www.metoffice.gov.uk, accessed on 29 July 2021) and utilized by the KMA for dynamical downscaling to seasonal and continental scales. It was used together with the Coordinated Regional Climate Downscaling Experiment [45] to prepare the national climate change scenario for South Korea. The HadGEM3-RA reproduces small-scale features, such as the coastline and intricate topography of the Korean Peninsula, more realistically than other GCMs because of their high resolution [45,46]. Therefore, we used the HadGEM3-RA to prepare the climate change scenarios (RCP 4.5 and RCP 8.5) using the “Dismo Package” in R [47], similar to previous studies performed in South Korea [48,49,50,51,52,53].

The current climate was determined from average climatic data recorded from 1950 to 2010, and the future climate temperatures for 2050 and 2070 were estimated from predictions for 2046 to 2055 and 2066 to 2075, respectively. Each climatic dataset had a spatial resolution of 30 arc seconds (~1 km^2^ at the equator), with the same spatial extent and geographical coordination system (WGS 1984 datum).

In addition to the bioclimatic variables, we used three other environmental variables, including seven categories of land cover (e.g., agricultural land, grassland, forest, urban area, water, wetland, and barren land), distance from roads (d-road), and distance from water (d-water) in the modeling of invasive weeds. The land cover change scenarios were downloaded from the Korea Adaptation Center for Climate Change (https://kaccc.kei.re.kr, accessed on 11 August 2021).

Roads and highways serve as ideal habitats and corridors for IIAPS to spread and proliferate. IIAPS that have become established along roads can serve as a source of invasion into the adjacent forests, grassland, and agricultural land [54]. South Korea is rich in water resources, with thousands of streams, rivers, and lakes across the country. IIAPS grow in the disturbed areas close to these water sources. Streams and rivers thus have an important role in IIAPS seed dispersal. Therefore, we prepared d-road and d-water in ArcGIS 10.3 (Esri, Redlands, CA, USA) using the Euclidian distance function with a resolution of 1 km^2^. To eliminate autocorrelation (r^2^ > 0.75, *p* = 0.05) among the environmental variables, we used Spearman’s correlation on pairs with the Proc Corr function of SAS 9.4 (SAS Institute, Inc., Cary, NC, USA). We picked nine important variables with low correlation and high predictive performance, as in Shin et al. [50] and Hong et al. [55] (Appendix A).

### 2.4. Species Distribution Modeling

Species distribution modeling (SDM) is an approach for estimating a species’ distribution throughout global space and time by applying a correlation between the species’ geographic occurrence and its surrounding environment [56]. The SDM approach has been used in various sectors of ecology over the last decade to predict species’ potential habitat under future climate changes [57], including IIAPS [49,58,59]. Various SDM tools such as machine learning, statistical regression, and spatial interpolation are currently used to model species distribution [56]. Among these tools, MaxEnt is a machine-learning modeling technique for predicting suitable habitats. It is used worldwide because it exhibits a high predictive performance from small sets of species presence data and environmental variables [60]. Thus, we performed MaxEnt modeling using “Biomod2” Package v.3.5.1, selecting a single model MAXENT.Phillips.2 [61] for the prediction of spatial distributions of ten selected IIAPS in South Korea. In this study, 75% of the species occurrence data were used for model calibration, and the remaining 25% were used for model validation. The MaxEnt model requires background points (e.g., pseudo-absences): we used ArcGIS 10.3 to determine 15,052 background points from the study area, as suggested by Barbet-Massin et al. [62]. The other options in the MaxEnt model were set to the default values, and the model was run 100 times.

### 2.5. Model Evaluation and Validation

To investigate the goodness-of-fit of the model used in this study, we examined three evaluating parameters: the area under the curve (AUC) values of the receiver operating characteristic (ROC) curves [63], the true skill statistic (TSS) [64], and the kappa statistic. The AUC, TSS, and kappa values were calculated from the test data points. The AUC is a threshold-independent technique for distinguishing presence from an absence that is used to test model outcomes. The AUC value ranges from 0 to 1 and evaluates the performance of a model [65].

The AUC value is independent of the data size (prevalence); however, its use can be criticized because it weights commission and omission errors equally and may avoid true predictions [66]. In particular, expanding the geographical range outside the occurrence range produces a high AUC value (overfitting), resulting in a misleading evaluation of the model [66]. The model was graded as follows, according to the AUC values: poor (0.6–0.7), fair (0.7–0.8), good (0.8–0.9), or excellent (0.9–1.0) [67]. The TSS accounts for both omission and commission errors, and it is used as an alternative criterion to authenticate model efficiency [64,68]. Similarly, the kappa value accesses the accuracy of prediction in comparison with what could have been gained by chance alone [64]. Both the TSS and kappa statistic range from −1 (poor agreement) to +1 (perfect prediction) [64]. We used all three parameters to confirm and validate model performance. Additionally, we performed a jackknife test to quantify the significance of the bioclimatic and environmental variables in the model performance. The binary habitat suitability maps obtained from MaxEnt modeling were used to assess the spatial distribution and spatial invasion of IIAPS. The database and detailed methodology are summarized in a flowchart in Figure 3.

### 2.6. Prediction of the Spatial Distribution of IIAPS

We estimated the areas of the current and future spatial distributions of each IIAPS and calculated the percentage changes for 2050 and 2070 (for both RCP 4.5 and RCP 8.5) relative to the current distribution area. To understand the differences in the spatial distribution and ordering of the samples [69], principal component analysis (PCA) was performed based on the predicted area covered by each IIAPS in the different administrative divisions of South Korea under current and future environmental conditions. The average spatial distributions of IIAPS in different groups were plotted with Map Algebra and the Spatial Analyst tool in ArcGIS 10.3 and compared between the different groups. Percentage changes in the average area for each group in 2050 and 2070 relative to the current area were calculated and expressed in graphical form.

### 2.7. Prediction of the Spatial Invasion Risk of IIAPS

The binary spatial distribution maps of the 10 IIAPS were summed with the Raster 3.4 package in GNU R4.03 to determine an aggregated map of spatial distribution under current and future (RCP 4.5 and RCP 8.5) environmental conditions. The aggregated spatial distribution map was used to estimate the invasion risk map, in which cells with higher species richness indicate high invasion risk and greater potential environmental problems. The current and future spatial invasion risks of the IIAPS were classified into three categories according to the level of species richness in each cell: low risk (<33%), moderate risk (>33% to 66%), and high risk (>66% to 100%). We used a linear scale and the method of Ahmad et al. [21], with minor modifications. The areas of low, moderate, and high invasion risk were estimated with the raster calculator in the Spatial Analyst tool in ArcGIS 10.3. We then calculated the area covered by each risk category relative to the country’s total area (under both RCP 4.5 and RCP 8.5). Similarly, we calculated the area covered by each risk category within an administrative division relative to the total area of that division, under both RCP 4.5 and 8.5. Moreover, to understand the invasion risk at the local level, we determined the average invasion risk map (under RCP 4.5 and RCP 8.5) for 250 districts using the zonal statistics in the Spatial Analyst tool in ArcGIS 10.3.

## 3. Results

### 3.1. Selection and Evaluation of Variables

To conduct the modeling proposed in this study and to select independent environmental variables, we measured Spearman’s correlations among three environmental variables and 19 bioclimatic variables (Appendix A). Six bioclimatic variables (annual mean temperature, isothermality, temperature seasonality, annual precipitation, precipitation in the wettest month, and precipitation in the driest month) and three environmental variables (distance from roads, distance from water, and land cover change) were ultimately selected for MaxEnt modeling based on their weak correlation with each other (r < 0.60; Appendix A). These nine variables were considered the most influential factors for the occurrence probability of IIAPS (Table 2).

Next, we determined the contribution of the nine variables to the MaxEnt model and assessed the degree of importance of each variable in the model. We measured the percent contribution of each variable to model performance using a heuristic approach [70]. In this approach, the contribution of a variable is estimated from the increase in model gain it provides [70]. Among the nine variables selected above, land cover change had the highest relative contribution for seven of the IIAPS [*Helianthus tuberosus* (69.41%), *Festuca arundinacea* (48.40%), *Coreopsis lanceolata* (44.77%), *Dactylis glomerata* (44.42%), *Amorpha fruticosa* (42.22%), *Poa pratensis* (41.67%), and *Medicago sativa* (35.37%)]. Similarly, Bio04, Bio12, and Bio13 were the most important variables for *Eragrostis curvula* (32.39%), *Lolium perenne* (37.50%), and *Ageratina altissima* (42.98%), respectively (Table 2). These results reveal that land cover change, temperature seasonality, annual precipitation, and precipitation in the wettest month were the most prominent driving factors for the species distribution models of the studied IIAPS; the other variables played minor roles in this study. We assessed variable importance using the jackknife approach [70], which measures how relevant each variable is in explaining species distribution and how much unique information each variable provides [71]. The jackknife test showed that five variables—annual mean temperature, temperature seasonality, annual precipitation, precipitation in the wettest month, and land cover change—were highly correlated in the model (Appendix A).

### 3.2. AUC, TSS, and Kappa Values Show Excellent Model Prediction for All IIAPS

The AUC, TSS, and kappa statistics assessed model performance, presented in Table 3. The average value of AUC was 0.767 ± 0.054, *n* = 10, ranging from 0.72 (*D. glomerata*) to 0.92 (*E. rugosum*), indicating that the proposed model was sufficiently accurate and that the outputs would be close to the real distribution. The ROC curves for each studied weed are presented in Appendix A. Similarly, the average value of TSS was 0.77 ± 0.03, *n* = 10, ranging between 0.72 (*F. arundinacea*) and 0.85 (*C*. *lanceolata*), and the mean kappa value was 0.678 ± 0.054, *n* = 10, ranging between 0.57 (*D. glomerata*) and 0.79 (*E. rugosum*). These AUC, TSS, and kappa values indicate that the observations may support the predictions provided by the model.

### 3.3. Environmental Changes Positively Regulate the Spatial Distribution of IIAPS in South Korea

We performed MaxEnt modeling of 10 IIAPS in South Korea, mapped the predicted spatial distributions of each plant species (Appendix A), and estimated the area (km^2^) covered by each plant species both currently and in the future (2050 and 2070), under climate change scenarios RCP 4.5 and RCP 8.5. Under the current environmental conditions, the average IIAPS spatial extent was 33,948 km^2^, covering 35.74% of the country’s total land surface; *M. sativa* had the highest coverage at 44,427 km^2^ (Table 4).

The model predicted that the spatial distribution of all IIAPS except *M. sativa* would increase in the future under RCP 4.5. The percentage changes in spatial distribution relative to the current distribution were estimated to range from −7% to 150% in 2050, and from −9% to 156% in 2070 (Table 4). Similarly, under RCP 8.5, the data show percentage changes in the spatial distributions of −10% to 107% in 2050 and from −16% to 145% in 2070 (Table 4). These results suggest that the spatial distribution of almost all of the tested IIAPS will have increased by both 2050 and 2070. Interestingly, the percentage change in spatial distribution was estimated to be lower under RCP 8.5 than RCP 4.5 for a number of the IIAPS (Table 4). This indicates that the spatial extent of IIAPS may be negatively affected by an extreme increase in global warming.

We performed PCA to determine the ordering of the IIAPS and identified three distinct groups (groups 1, 2, and 3) having similar spatial distributions and spread potential (Figure 4). The IIAPS present in group 1 (*M. sativa*), group 2 (*A*. *fruticose*, *D*. *glomerate*, *F*. *arundinacea*, *H*. *tuberosus*, and *P*. *pratensis*), and group 3 (*C*. *lanceolate*, *E*. *curvula*, *E*. *rugosum*, and *L*. *perenne*) had small, intermediate, and large distribution areas, respectively. The average spatial distributions of the IIAPS in each group are presented in Appendix A. We calculated the percentage change in spatial coverage for each group under RCP 4.5 (Figure 5A) and RCP 8.5 (Figure 5B) relative to the current coverage. We found that group 3 had the highest increase in 2050 (increases up to 92.8%) and in 2070 (increases up to 102%). Group 1 had the lowest percentage change with estimated reductions in the average spatial distribution in 2050 (up to −10%) and 2070 (up to −16%).

### 3.4. Environmental Changes Increase the Spatial Invasion Risk of IIAPS in South Korea

Invasion risk estimates reveal that a non-native species can become established in a novel ecosystem, either deliberately or haphazardly, thereby threatening the surrounding native biodiversity. Here, we estimated the invasion risk of IIAPS based on the number of introduced alien plant species present in a cell: cells with a higher number of IIAPS were considered to be at a higher risk of invasion. We classified the results into three categories of risk (low, moderate, and high). The overall invasion risk maps calculated from the 10 IIAPS under current and future (RCP 4.5 and RCP 8.5) environmental conditions are presented in Figure 6.

Under current conditions, the risk of invasion was predicted to be high in three provinces [Gyeonggi (e.g., Gimpo-si and Ilsanseo-gu of Goyang-si), Jeollabuk (e.g., Gimje-si and Iksan-si), and Jeollanam (e.g., Mokpo-si)] and four metropolitan cities [Seoul (e.g., Seongdong-gu and Dongdaemun-gu), Busan (e.g., Gangseo-gu), Ulsan (e.g., Nam-gu), and Gwangju (e.g., Seo-gu)] (Figure 6A and Figure 7A), covering an estimated 31.51–70.31% of the corresponding provinces and cities (Table 5). However, the risk of invasion was estimated to be low to moderate in the northern part of Gyeonggi province and across most of the Gangwon and Jeju provinces (Figure 6A).

Future environmental change increased the extent and intensity of the IIAPS invasion in South Korea for both 2050 and 2070 (Figure 6B,C). The model predicted that most IIAPS would retain the current spatial distribution and add significant additional suitable habitat; therefore, the risk of invasion would be moderate to high across most of Gyeongsangnam province, Gyeongsangbuk province, Chungcheongbuk province, central parts of Chungcheongnam province, three metropolitan cities (Daegu, Daejeon, and Sejong), and southern parts of Gangwon province (Figure 6B,C and Figure 7B,C). These provinces and metropolitan cities would have a high risk of invasion (up to 79.15%) by 2050 and 2070 (Table 5). These results indicate that future environmental changes may lead to expanded spatial invasion of IIAPS from the south coast and west coast areas toward the central and northern regions.

Under current conditions, the proportions of areas estimated to be at low, moderate, and high risk comprise 47.96%, 22.88%, and 29.14% of the country’s total land mass, respectively (Figure 8A). Our study predicts that 82.36% and 86.21% of the country’s total area may be at moderate or high risk of invasion by 2050 and 2070, respectively, under RCP 4.5 (Figure 8A). Similarly, under RCP 8.5, 81.31% and 80.99% of the country’s total area may be at moderate or high risk of IIAPS invasion by 2050 and 2070, respectively (Figure 8B). These results show that the rate of spatial invasion by IIAPS may increase significantly in South Korea with future environmental changes.

## 4. Discussion

Our study provides several prominent findings, as follows: (1) Of the nine environmental variables used in the model, land cover change was the most important for the future spatial distribution of IIAPS (Table 2). (2) Under future scenarios of environmental change in South Korea, alien plant species introduced for a particular purpose will not remain limited to the introduced area but will spread and invade non-targeted ecosystems, including croplands, pasture, and forest. (3) Current invasion risk is estimated to be high in the coastal areas and some densely populated metropolitan cities (e.g., Seoul, Busan, and Gwangju; Figure 6A) but, in the future, invasion risk is predicted to increase in the central and northern regions, leading to approximately 86.21% of the total area of the country being at high or moderate risk of invasion (Figure 6B,C). (4) Extreme climate change may not be favorable overall to alien and invasive species: the relative change in IIAPS coverage was generally lower under climate change scenario RCP 8.5 than under RCP 4.5 (Table 4).

Of the total number of alien and invasive plant species recorded in South Korea, 25.31% were intentionally introduced into the country [27]. The alien plants have been intentionally introduced into South Korea for a variety of purposes, including erosion control in the mountains (e.g., *A*. *fruticose*, *C*. *lanceolate*, and *E*. *curvula*), grassland management for commercial livestock farming (e.g., *M*. *sativa*, *F*. *arundinacea*, and *L*. *perenne*), ornamental use (e.g., *E. rugosum)*, and medical use (e.g., *H*. *tuberosus*; Table 1) [27]. These species can become well adapted, cultivated, and naturalized in local and systemic areas [13,20,55,72]. Some now exist as ruderal species colonizing disturbed land, such as roads, shorelines, mining sites, and recreational parks [27,29]. Ironically, the ecological niches of these species trigger their negative effects on agriculture, horticulture, native ecosystems, the economy, crop production, livestock, and wild ungulates [73]. *E*. *rugosum* is now listed as the most ecologically disruptive weed in South Korea [27], and other species may become serious threats to the natural ecosystem. With these phenomena on the rise in South Korea, the issues of how to monitor IIAPS, predict their spatial distribution and manage invasion risk are being raised. Therefore, modern modeling systems are now an essential tool for protecting the natural ecosystem.

MaxEnt is a presence-only modeling approach that is suitable for cases in which true absence data is lacking. It is commonly used for modeling invasive species because such species’ ranges are increasing and have not yet reached equilibrium; the absence of data for invasive species is therefore untrustworthy and may lead to incorrect interpretation [32,33]. MaxEnt is a generative strategy that uses environmental data from across the study area rather than a discriminative approach and has the advantage of being more efficient when presence data are not sufficient [74]. The MaxEnt model produces robust estimates of climatically suitable habitats for invasive species at small spatial scales and with a limited dataset [75,76]. Although recent developments are attempting to solve some of the limitations of MaxEnt modeling, major drawbacks remain, including the risk of overfitting, which limits the model’s ability to generalize well to new data. The ’regularization multiplier’ parameter in MaxEnt attempts to solve this by reducing the model’s complexity, resulting in a less localized prediction [32]. Another important drawback to MaxEnt modeling is the accuracy of presence-only modeling related to biases in the species occurrence localities. This study spatially rarefied the presence points using the Arc GIS SDM toolbox 2.4 to reduce species occurrence biases.

In addition to MaxEnt, the most popular machine-learning models for studying species distribution are artificial neural networks (ANNs), random forests (RFs), and genetic algorithms for rule-set production (GARPs). These models are powerful tools for solving complex dependencies, and both numerical and categorical environmental variables can be used as input variables in such models [77]. An ANN consists of a network of artificial neurons or nodes, and all information transfer between neurons is weighted. ANNs can operate like multiple regressions, and their accuracy is controlled by the weight decay of the links and the number of hidden neurons [77]. They show good performance with complicated species–environment relationships. RFs are multi-decision tree ensemble classifiers that use Breiman’s random forest algorithm for classification and regression [78]. They are unaffected by multicollinearity, excel at dealing with correlated variables, and run efficiently with large databases [76,78]. Similarly, GARPs are SDMs that use a genetic algorithm to create random mathematical rules that can be interpreted as limiting environmental circumstances and specific species–environment correlations [77,79]. Each rule is treated as a gene and sets of genes are combined at random to build a large number of models reflecting the possibility of species occurrence [77]. Although all the models described above have their own unique properties for predicting ecological niches, MaxEnt is considered an effective SDM tool for robust predictions across many species and regions. It is also relatively user-friendly as the software can import GIS layers of environmental variables directly using presence-only species occurrence data with default settings; less effort needs to be paid to parameter tuning [32,33,74].

Because of the current situation, we attempted to develop a modeling system that could monitor and estimate the spatial distribution and invasion risk of 10 IIAPS using MaxEnt. The results revealed that all IIAPS in our study except *M. sativa* would retain their current ecological niches and add additional areas to their spatial distribution in South Korea. However, the rate and extent of increasing coverage are not estimated to be consistent among all IIAPS. *D. glomerata*, *E*. *curvula*, and *E*. *rugosum* are estimated to have relatively high spatial coverage, up to 80.85% and 77.10% of the total land area of South Korea in 2050 and 2070, respectively (under RCP 4.5). These predictions are comparable to the results from earlier studies performed in South Korea [49,55,80]. Similarly, we estimated the risk of invasion by the IIAPS and found that 12% of the country is currently at high risk of invasion and that this area is likely to increase dramatically by 2050 (to 39%) and 2070 (to 50%), under RCP 4.5. For all IIAPS, the increased invasion risk with environmental change expanded in eastward and northward directions, invading major agricultural pocket zones (e.g., Jeollabuk, Jeollanam, and Chungcheongnam provinces), most of the national parks, and protected areas, including the Baekdudaegan mountain range (701 km) [48], which are biodiversity hotspots and the natural habitat of 297 species of endemic plants (e.g., *Abies koreana*, *Berberis koreana*, and *Arabis columnaris*) [81,82]. Moreover, global climate change is likely to increase the capacity of alien plant species to invade into new areas while lowering native community resistance to invasion by disrupting the dynamic equilibria that maintain native communities [16]. Among the various climatic variables, temperature seasonality and annual precipitation are the dominant factors affecting the spatial distribution of many IIAPS (Table 2).

The results supported by many other studies have similarly attempted to model the invasion risk of non-native species in South Korea and across the globe. For example, Dullinger et al. [83] estimated that future climate and land cover changes in Europe would increase the risk of naturalization of non-native garden plants by up to 102% under RCP 8.5. Bai et al. conducted a spatial invasion risk assessment of alien invasive plants in China and found that the southern part of China will be at high risk of invasion in the future, with drought-resistant species becoming dominant in the natural ecosystems. In the eastern United States, Bradley et al. [13] estimated the invasion risk of three invasive plants, *Pueraria lobata*, *Ligustrum sinense*, and *Imperata cylindrical*. They showed that climate change is likely to allow significant expansion of their ranges. Similarly, risk assessment of the invasive weed Leucanthemum vulgare under future climate change showed it would be distributed in all continents, with the Oceania region being at particularly high risk [21]. Adhikari et al. [49] and Hong et al. [55] estimated rapid range expansion of invasive species in South Korea under future climate change. These studies demonstrate that future climate change will be a major factor in the proliferation of non-native plants across the world, consistent with our findings.

Plant invasion is thought to be boosted by soil disturbance induced by anthropogenic land cover changes, which accelerate ecosystem disruption and favor introducing invasive species over native species [84]. In South Korea, land use patterns have been changing for a long time. Specific examples of major land cover changes in South Korea include the expansion of road and rail networks, implementation of upland farming practices, forest fragmentation for urbanization, and increased industrialization [30]. Although the transportation network is part of the fundamental infrastructure of the country, without careful design and management of the roads, they may be a source for the introduction and dispersal of alien plant species into new areas [25,85]. In South Korea, the total length of roads and railways that are expected to increase plant invasion has been measured at 116,850.6 km [86]. Understanding the impact of land cover changes on ecological niche availability is crucial to prognosticating invasion and managing landscapes to minimize the spread of invasive species [24,25]. Currently, the spatial distribution of these species is concentrated close to the coastal areas in the western and southern regions: we predict that the distributions will expand toward the central and northern regions by 2050 and 2070, related to the transportation corridors to these regions. Thus, land cover changes and climatic variables may play pivotal roles in the spatial distribution of IIAPS, as suggested by the modeling system used in this study and by previous research [49,55].

Our findings provide spatially explicit evidence that supports the earlier hypothesis that warming temperatures will increase the northward spread of alien and invasive plant habitats [87]. These results demonstrate that future environmental changes, including climate change and land cover change, are likely to favor IIAPS in South Korea. Therefore, the future spatial distribution and invasion risk of IIAPS will be exacerbated on a large scale, threatening the native biodiversity and imperiling the native ecosystem of South Korea [13,55]. In addition to the environmental factors described in this study, the introduction history, biogeographic origin, and biological traits of IIAPS are strongly correlated with invasion success [88]. The biological traits that are correlated with invasiveness include relative growth rate [89], seed mass, the maximum height of the plant, and plasticity [90]. Plants with shorter life cycles usually have higher reproduction rates and may evolve more quickly to adapt to new environments, improving invasion success [91]. Similarly, the physiological characteristics of alien plants, such as their photosynthetic rate, resource utilization efficiency, and tolerance to environments from humid to xeric, play important roles in their invasion success [88]. Therefore, biological and physiological traits should be incorporated during the modeling of IIAPS to obtain the most accurate model predictions. Other important parameters for prediction include the soil characteristics, land topography, biotic interactions (e.g., competition and facilitation), and vectors driving species invasion, as suggested by Buri et al. [92] and Pysek and Richardson [93]. Although diverse variables were included in our model to improve accuracy, these other variables should be analyzed further in future studies to obtain more precise predictions.

## 5. Conclusions

IIAPS can negatively affect a variety of fields in South Korea, including agriculture, the economy, industry, horticulture, and the natural ecosystem. As part of efforts to overcome this, we developed a MaxEnt prediction model to estimate the spatial distribution and invasion risk of IIAPS under current and future environmental changes. Our findings suggest that the spatial distributions of IIAPS are estimated to enlarge extensively in the future while retaining their existing ecological niches, and that, currently, the southern and western coastal regions and some metropolitan cities, such as Seoul, Busan, and Daegu, are at relatively high risk of invasion by IIAPS. In addition, climate change due to global warming and other environmental changes is likely to increase invasion risk in the country. Taken together, our results strongly suggest that this modeling system can help to prioritize the invasive weeds and geographical regions to be targeted in a timely fashion, as well as support government authorities in adopting the best preventive measures. For example, our system can be used to support efforts to eradicate small populations of IIAPS detected in non-target areas that are likely to become sources for future expansion.

## Figures and Tables

**Figure 1 biology-10-01169-f001:**
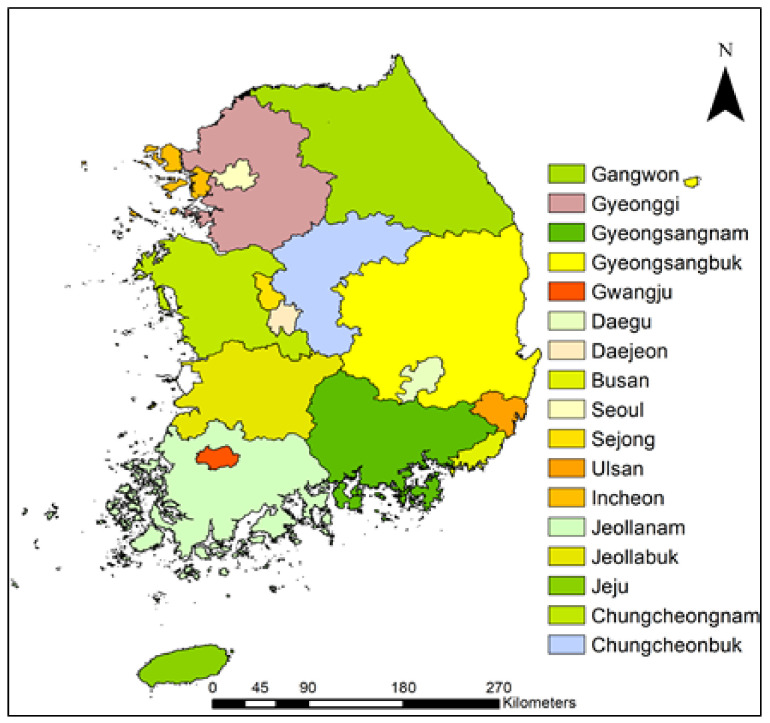
Administrative divisions (major cities and provinces) and geographical boundaries of South Korea.

**Figure 2 biology-10-01169-f002:**
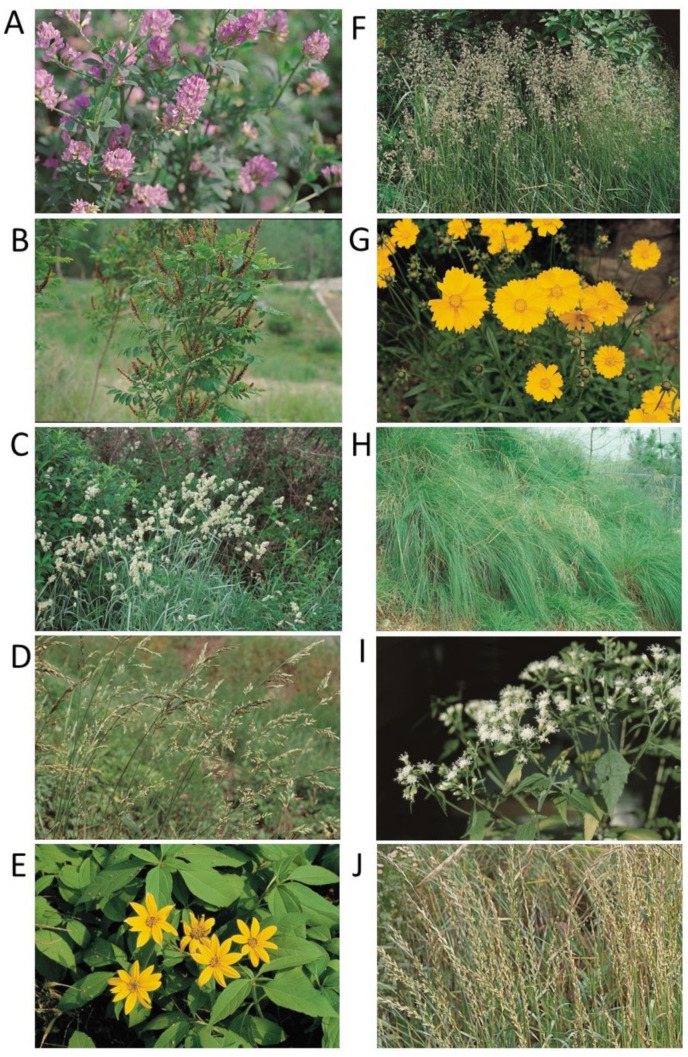
Photographs of ten introduced alien plant species in South Korea. (**A**) *Medicago sativa*; (**B**) *Amorpha fruticose*; (**C**) *Dactylis glomerate*; (**D**) *Festuca arundinacea*; (**E**) *Helianthus tuberosus*; (**F**) *Poa pratensis*; (**G**) *Coreopsis lanceolate*; (**H**) *Eragrostis curvula*; (**I**) *Ageratina altissima*; and (**J**) *Lolium perenne*.

**Figure 3 biology-10-01169-f003:**
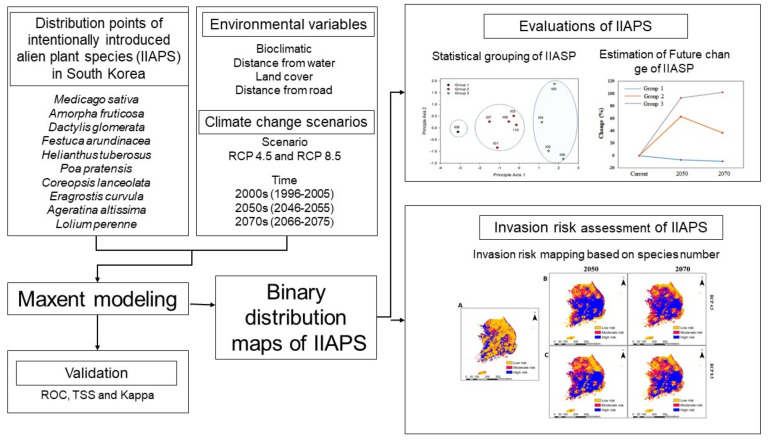
Flow—chart of the development and building blocks of the MaxEnt model and its practical application in estimating the invasion risk of intentionally introduced alien plant species in South Korea.

**Figure 4 biology-10-01169-f004:**
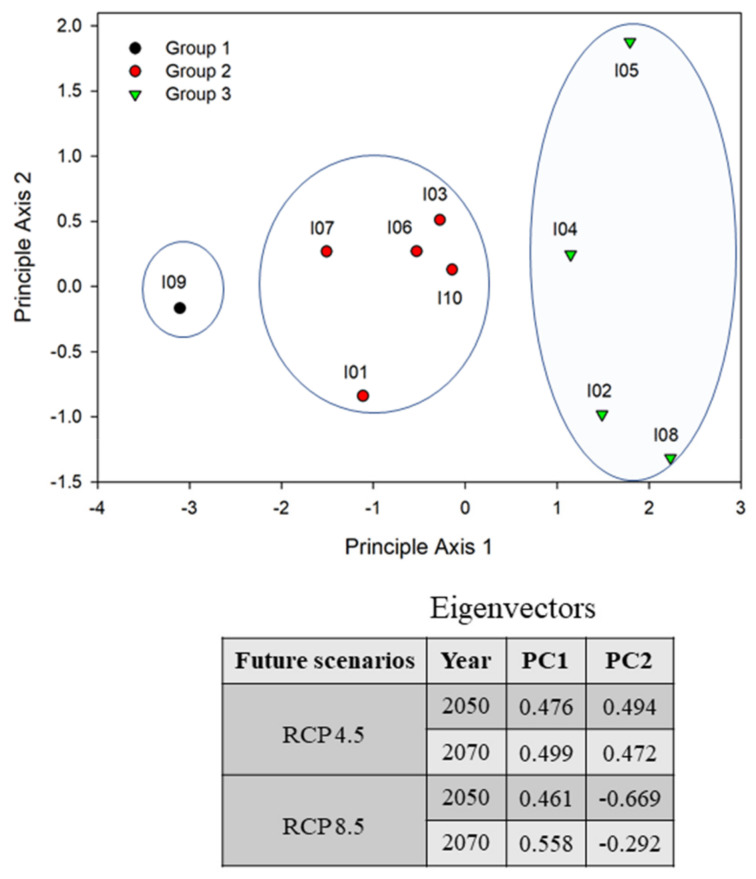
Principal component analysis reveals the grouping of introduced alien plant species based on the extent of their spatial distribution. The scientific and common names of the introduced alien plants (I01−I10) are presented in Table 1.

**Figure 5 biology-10-01169-f005:**
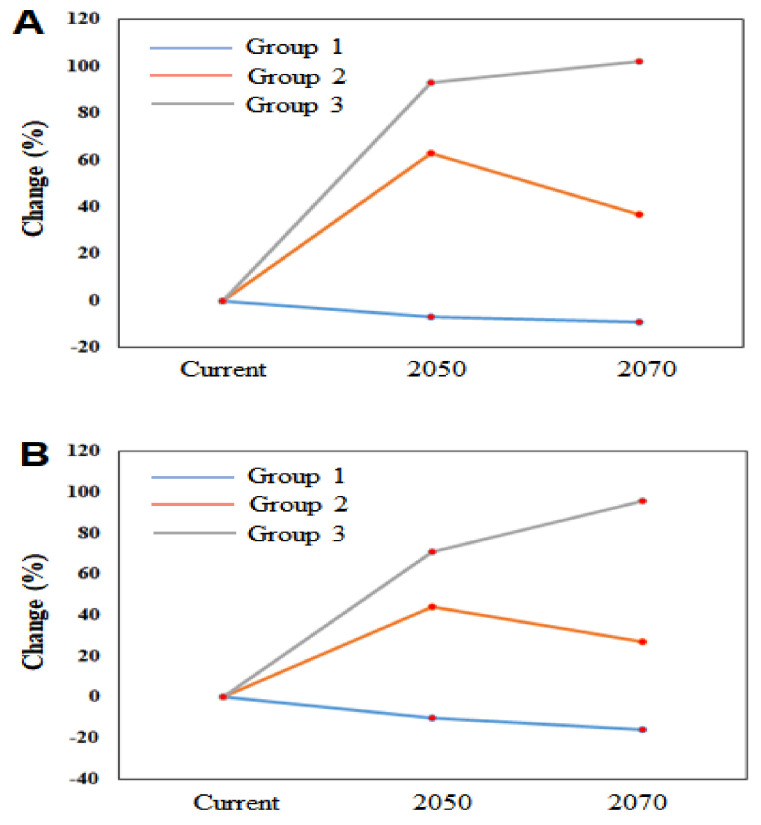
Percentage change in the extent of the average spatial distribution of alien plant species in group 1, group 2, and group 3, under the climate change scenarios RCP 4.5 (**A**) and RCP 8.5 (**B**). The details of each group are presented in Table 1.

**Figure 6 biology-10-01169-f006:**
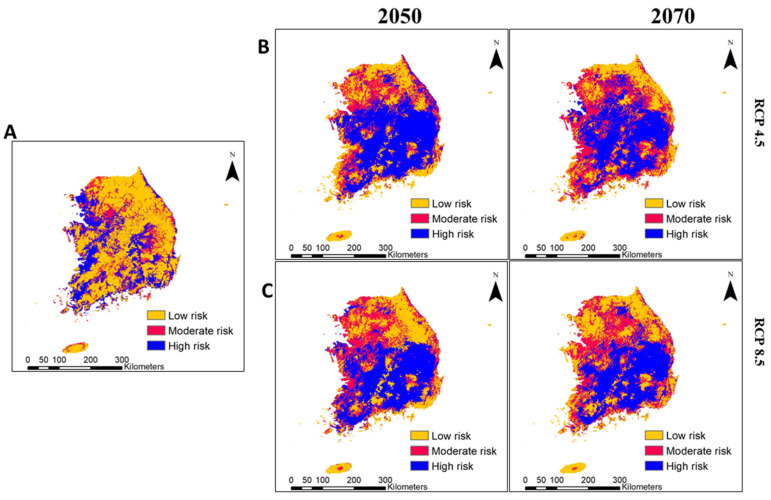
Predicted invasion risk of introduced alien plants in South Korea. The invasion risk assessment is divided into three categories: low, moderate, and high. These three categories are indicated by yellow, red, and blue in the figure. Spatial invasion predictions under the current (**A**) and future (**B**,**C**) environmental conditions in South Korea.

**Figure 7 biology-10-01169-f007:**
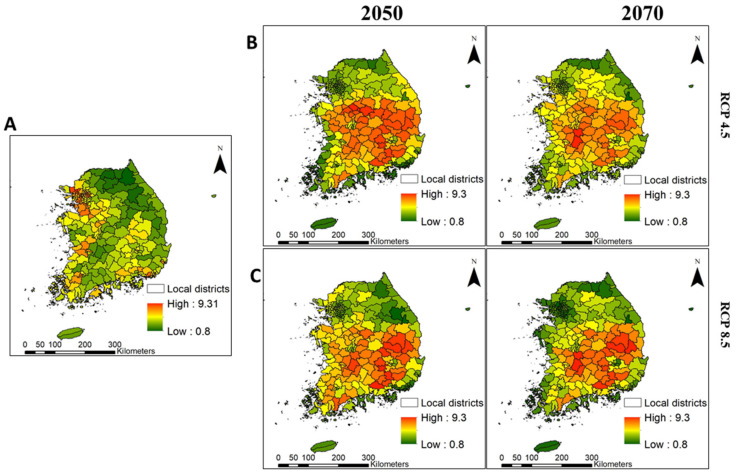
Average invasion risk at the district level under the current (**A**) and future (**B**,**C**) environmental conditions in South Korea. The red colors in the figure indicate the highest average invasion risk, and the green colors indicate the lowest average invasion risk.

**Figure 8 biology-10-01169-f008:**
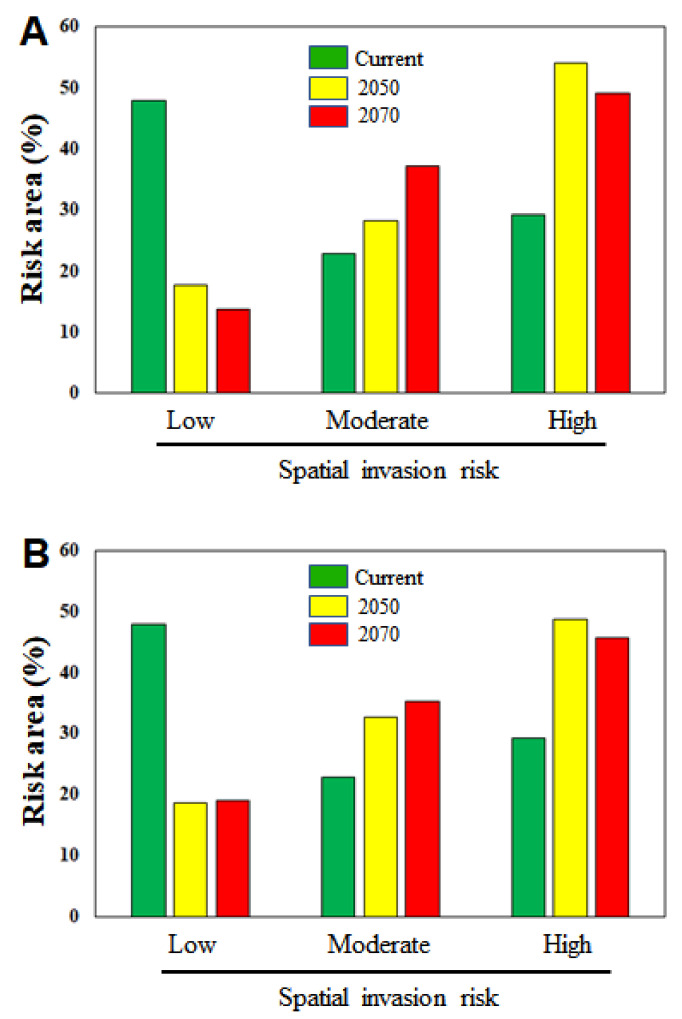
Estimates of the area assigned to different categories of invasion risk relative to the total area of South Korea. The numbers on the *y*-axis indicate the percentage area categorized as at low, moderate, or high risk of invasion under current and future environmental conditions. (**A**) RCP 4.5 and (**B**) RCP 8.5.

**Table 1 biology-10-01169-t001:** List of intentionally introduced alien plant species (IIAPS) used in the species distribution model.

IAPAS Group ^a^	ID No.	Scientific Name	Common Name	Native Range	Mode of Introduction	Introduction Period	Degree of Naturalization
Group 1	I223	*Medicago sativa*	Alfalfa	Mediterranean	Intentional (Pasture)	Before 1949	III
Group 2	I072	*Amorpha fruticosa*	Bastard indigo bush	North America	Intentional (Erosion control)	Before 1949	V
	I138	*Dactylis glomerata*	Orchard grass	North Africa	Intentional (Pasture)	Before 1949	V
	I165	*Festuca arundinacea*	Tall fescue	North Africa	Intentional (Pasture)	Before 1970	V
	I176	*Helianthus tuberosus*	Jerusalem artichoke	North America	Intentional (medicinal)	Before 1911	V
	I258	*Poa pratensis*	Kentucky bluegrass	Temperate zone	Intentional (Erosion control)	Before 1949	IV
Group 3	I129	*Coreopsis lanceolata*	Lance leaf coreopsis	North America	Intentional (Erosion control)	Before 1963	V
	I150	*Eragrostis curvula*	African love grass	North Africa	Intentional (Erosion control)	Before 1990	IV
	I157	*Ageratina altissima*	White snakeroot	North America	Intentional (Gardening)	Before 1990	IV
	I210	*Lolium perenne*	Ryegrass	North Africa	Intentional (Pasture)	Before 1970	IV

^a^ Division into group 1, group 2, or group 3 was based on the principal component analysis of the spatial distribution of intentionally introduced alien plant species (IIAPS) shown in Figure 3. The Roman numerals I to IV in the last column indicate the degree of naturalization of the IIAPS. I, rarely; II, low density and distributed in a small area; III, low density but distributed widely; IV, high density and distributed locally; V, high density and distributed widely.

**Table 2 biology-10-01169-t002:** Contribution of bioclimatic and environmental variables in the model.

Name of Species	Bio1	Bio3	Bio4	Bio12	Bio13	Bio14	d-Road	d-Water	Land Cover
*Amorpha fruticosa*	6.15 ^a^	10.73	6.72	8.54	0.76	11.28	4.09	9.51	42.22
*Coreopsis lanceolata*	8.50	4.93	10.03	7.98	3.88	2.23	8.27	9.41	44.77
*Dactylis glomerata*	6.03	5.59	12.62	0.67	20.79	5.74	3.06	1.07	44.42
*Eragrostis curvula*	23.67	13.53	32.39	2.07	8.48	6.23	1.70	0.00	11.93
*Ageratina altissima*	4.18	18.88	12.72	3.37	42.98	11.10	4.94	0.42	1.42
*Festuca arundinacea*	16.33	11.51	0.88	6.44	8.33	3.77	2.33	2.01	48.40
*Helianthus tuberosus*	8.67	1.50	12.37	2.28	0.42	0.87	2.00	2.49	69.41
*Lolium perenne*	11.83	1.15	7.71	37.50	3.72	2.40	3.73	0.52	31.42
*Medicago sativa*	1.15	0.68	15.34	34.30	11.17	1.66	0.16	0.17	35.37
*Poa pratensis*	2.05	11.51	5.51	14.09	8.43	14.06	1.65	1.03	41.67

^a^ Percentage contribution. The variables Bio1, Bio3, Bio4, Bio12, Bio13, and Bio14 indicate six bioclimatic variables: annual mean temperature, isothermality, temperature seasonality, annual precipitation, precipitation in the wettest month, and precipitation in the driest month, respectively. Similarly, the variables d-road, d-water, and land cover indicate three environmental variables: distance from roads, distance from water, and land cover change, respectively.

**Table 3 biology-10-01169-t003:** AUC, TSS, and kappa values for each IIAPS in the model calibration.

Name of Species	No. of Species Presence Points	AUC Value	TSS Value	Kappa Value
*Amorpha fruticosa*	516	0.76	0.79	0.67
*Coreopsis lanceolata*	806	0.73	0.85	0.66
*Dactylis glomerata*	634	0.72	0.81	0.57
*Eragrostis curvula*	110	0.75	0.77	0.67
*Ageratina altissima*	104	0.92	0.77	0.79
*Festuca arundinacea*	1076	0.73	0.72	0.64
*Helianthus tuberosus*	734	0.74	0.75	0.71
*Lolium perenne*	228	0.78	0.76	0.66
*Medicago sativa*	242	0.76	0.74	0.68

**Table 4 biology-10-01169-t004:** Relative change in the spatial distribution of IIAPS in South Korea.

Species Names	Current (km^2^)	RCP 4.5		RCP 8.5	
		2050 (%)	2070 (%)	2050 (%)	2070 (%)
*Medicago sativa*	44,427	−7	−9	−10	−16
*Amorpha fruticosa*	38,060	43	8	63	23
*Dactylis glomerata*	37,565	61	91	42	37
*Festuca arundinacea*	32,317	83	34	39	45
*Helianthus tuberosus*	38,656	65	14	33	2
*Poa pratensis*	34,272	41	94	42	66
*Coreopsis lanceolata*	30,027	98	64	101	98
*Eragrostis curvula*	38,113	101	92	59	98
*Ageratina altissima*	15,725	150	156	45	71
*Lolium perenne*	30,317	74	104	107	145

**Table 5 biology-10-01169-t005:** Area in each invasion risk category relative to the total area of the province or major city, under current and predicted environmental changes in South Korea.

Provinces	Total Area (Km^2^) ^a^	Current (%) ^b^			2050 (%) ^c^			2070 (%) ^d^		
		Low	Moderate	High	Low	Moderate	High	Low	Moderate	High
Gangwon	16,503.73	72.21	20.62	6.94	36.78	45.29	17.93	36.15	41.67	22.18
Gyeonggi	9810.10	30.64	33.41	35.68	16.34	60.51	23.14	21.34	53.88	24.78
Incheon	614.89	17.27	34.32	45.63	16.53	71.34	12.13	23.68	63.59	12.74
Seoul	605.70	6.18	23.51	70.31	21.95	75.84	2.21	32.75	67.09	0.16
Gyeongsangbuk	18,922.94	48.49	23.36	28.13	6.73	14.12	79.15	5.73	22.07	72.20
Chungcheongbuk	7415.68	50.23	20.74	29.02	8.09	18.47	73.44	4.73	23.22	72.05
Chungcheongnam	7637.76	39.75	20.54	39.60	8.20	23.58	68.22	11.81	35.51	52.69
Sejong	465.24	21.33	21.74	56.93	2.77	18.53	78.70	5.27	30.39	64.35
Daejeon	539.55	38.72	19.04	42.24	19.17	52.54	28.28	9.71	44.62	45.67
Jeollabuk	7716.82	47.17	16.15	36.67	9.28	22.07	68.65	6.25	26.89	66.87
Daegu	880.84	46.35	16.02	37.63	24.38	31.27	44.34	26.64	45.33	28.03
Gyeongsangnam	9809.70	39.45	25.01	35.46	18.30	17.55	64.15	7.53	32.56	59.92
Ulsan	1029.49	40.58	27.72	31.51	27.03	42.72	30.25	19.73	58.85	21.42
Jeollanam	10,180.34	46.09	17.38	36.24	20.79	32.52	46.69	18.59	47.81	33.60
Busan	673.03	29.86	16.86	53.28	80.23	18.14	1.63	16.04	76.83	7.13
Gwangju	498.36	24.85	10.77	64.37	19.74	29.56	50.70	19.17	34.78	46.05
Jeju	1674.96	45.33	54.26	0.18	57.71	42.29	0.00	59.84	40.15	0.00

^a^, Estimated area of the different provinces and major cities in South Korea. ^b,c,d^, The average percentage across the results for RCP 4.5 and RCP 8.5.

## Data Availability

Not applicable.

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
