# Peer review of "Assessment of the Spatial Invasion Risk of Intentionally Introduced Alien Plant Species (IIAPS) under Environmental Change in South Korea"

_biology, 2021, doi:10.3390/biology10111169_

Round 1

Reviewer 1 Report

Both Simple summary and Abstract describe the study in informative and compact way. Introduction focuses to the problem and its background present them in a clear form. The aim of the study is presented but in quite general manner (could be written in the form of short questions).

Material and methods describes thoroughly the study area and climate conditions. Technical details of the analyses are also well presented. The species chosen to the study are described very shortly mainly in a table form but reason to select these species is given. The methods seem to be well thought and data used is convincing, so there is no reason the doubt the results.

Results are clearly presented. All tables are, however, not understandable as such. E.g. Table 2 presents the variables as Bio1, Bio3, etc. which means that reader must search these from the text to understand the message. Using short title of the variable would this be better.

In discussion, authors highlight the most important findings and also describe several possible reasons for future invasions. It was interesting to see that there was no big difference among RCPs, and it would have been nice to see if the maps were still similar if the effects climate change was not taken into account. The impact of land use was highlighted, but its importance could be discussed even more. It would have been interesting to read more about the ecology of species and how it can be seen in the results. But as the paper is already pretty long, lacking of these is understandable.

This MS has clear structure and it is written in pleasant and fluent English. There are only small things in finishing (e.g. width of tables) that need some editing.

Author Response

  1. Both Simple summary and Abstract describe the study in informative and compact way. Introduction focuses to the problem and its background present them in a clear form. The aim of the study is presented but in quite general manner (could be written in the form of short questions).

Response: Thanks for the great comment. As you suggested, we added it in introduction part (lines 86-102).

  1. Material and methods describes thoroughly the study area and climate conditions. Technical details of the analyses are also well presented. The species chosen to the study are described very shortly mainly in a table form but reason to select these species is given. The methods seem to be well thought and data used is convincing, so there is no reason the doubt the results. Results are clearly presented. All tables are, however, not understandable as such. E.g. Table 2 presents the variables as Bio1, Bio3, etc. which means that reader must search these from the text to understand the message. Using short title of the variable would this be better.

Response: Thanks for the great comment. As you suggested, we wrote short description of all variables in Table 2 (lines 320-323).

  1. In discussion, authors highlight the most important findings and also describe several possible reasons for future invasions. It was interesting to see that there was no big difference among RCPs, and it would have been nice to see if the maps were still similar if the effects climate change was not taken into account. The impact of land use was highlighted, but its importance could be discussed even more.

Response: Thanks for the great comment. As you suggested, we wrote and added the impact of land cover change in discussion (lines 548-560).

  1. It would have been interesting to read more about the ecology of species and how it can be seen in the results. But as the paper is already pretty long, lacking of these is understandable.

Response: Regarding length of the paper, we agreed with your opinion. Although we tried to the our best to reduce volume of paper and to re-arrange and modify the texts based on the flow of logical story, it was still relatively long because the other reviewer wanted to add more information in discussion part. Please understand our situation during revision process. Again, we appreciate for your excellent suggestion.

Reviewer 2 Report

Comment and suggestion to authors:

Manuscript ID: biology-1399000

Titled:  "Assessment of the spatial invasion risk of intentionally intro-duced alien plant species (IIAPS) under environmental change in South Korea"

  1. In the section “1. Introduction”, the short and clear objectives of this study should be presented.
  2. There are a large number of introduced alien plant species, the authors should clearly explain what are the criteria to select the 10 species in this study.
  3. According to the list of 10 scientific names in the table 1 - 4, some of them is not the accepted name, it is a synonym of the other species. The authors should be very carefully check and correct this point.
  4. The photo of the 10 intentionally introduced alien plant species (IIAPS) and their invasion in South Korea should be added into this manuscript, so as to show the readers about the invasion risk of these plants in South Korea.
  5. Both pro and con of the MaxEnt prediction model should be discussed and compare with the other related models.
  6. The additional previous related publications related to this study should be add to compare and discuss with the presented results from this study.
  7. There are some spelling mistakes and grammatical error found in this manuscript, the author should check the whole manuscript before re-submission.

Author Response

Reviewer 2

1.In the section “1. Introduction”, the short and clear objectives of this study should be presented.

Response: Thanks for the great comment. As you suggested, we added it in introduction part (lines 86-102).

2.There are a large number of introduced alien plant species, the authors should clearly explain what are the criteria to select the 10 species in this study.

Response: It was an excellent comment. Ten alien plant species introduced intentionally in South Korea were selected based on the basis of their rapid range expansion and most invading, their degree of invasion into natural ecosystems, and the availability of minimum species occurrence records. These notions were added in the main texts (lines 134-137).

3.According to the list of 10 scientific names in the table 1 - 4, some of them is not the accepted name, it is a synonym of the other species. The authors should be very carefully check and correct this point.

Response: We agreed with your suggestion. After carefully checking this out, the scientific name of white snakeroot was corrected to Ageratina altissima and added to the table 1-4 and Fig.3. Thanks for the great comment.

4.The photo of the 10 intentionally introduced alien plant species (IIAPS) and their invasion in South Korea should be added into this manuscript, so as to show the readers about the invasion risk of these plants in South Korea.

Response: We agreed with your suggestion. We made a new Figure 2 that contained all photos (lines 179-184).

5.Both pro and con of the MaxEnt prediction model should be discussed and compare with the other related models.

Response: Thanks for the great comment. As you suggested, we added both pro and con of the MaxEnt prediction (lines 473-488) and added comparation of MaxEnt with machine learning models that were closed to MaxEnt (lines 489-509).

6.The additional previous related publications related to this study should be add to compare and discuss with the presented results from this study.

Response: As you suggested, we added contents as compared with previous works (lines 532-547).

7.There are some spelling mistakes and grammatical error found in this manuscript, the author should check the whole manuscript before re-submission.

Response: As you suggested, we checked these out and sent the paper to Bioedit LTD (www. Bioedit.com, job number: 26256). Thanks for the good comment.

Round 2

Reviewer 2 Report

Some spelling mistakes and grammatical error are remain in this manuscript, the authors should carefully check again.